# Efficacy and Long-Term Safety of Ibuprofen Gargle for Oral Lichen Planus: A Study Protocol of Randomized Crossover and Long-Term Extension Trials

**DOI:** 10.3390/mps6010007

**Published:** 2023-01-10

**Authors:** Yumi Kitahiro, Takeshi Ioroi, Yasumasa Kakei, Junya Yamashita, Akira Kimoto, Takumi Hasegawa, Asami Morioka, Kazuhiro Yamamoto, Masaya Akashi, Ikuko Yano

**Affiliations:** 1Department of Pharmacy, Kobe University Hospital, 7-5-2 Kusunoki-cho, Kobe 650-0017, Japan; 2Department of Oral and Maxillofacial Surgery, Kobe University Graduate School of Medicine, 7-5-2 Kusunoki-cho, Kobe 650-0017, Japan

**Keywords:** ibuprofen, gargle, oral lichen planus, one-arm clinical trial, randomized crossover controlled trial, pain, long-term safety

## Abstract

Oral lichen planus (OLP) is a type of chronic and refractory stomatitis characterized by abnormal keratinization, which is often painful. There is no consensus regarding treatment options for OLP, particularly in the presence of pain. The current study protocol focuses on the short-term efficacy and long-term safety of an ibuprofen gargle for pain management in patients with OLP. Patients (*n* = 24) with painful OLP will be enrolled. During a crossover study period, patients in the ibuprofen–placebo (IP) group will receive an ibuprofen gargle (0.6%) on day 1, a placebo gargle on day 2, and an ibuprofen gargle on days 3–5 at least once daily. Patients in the placebo–ibuprofen (PI) group will receive a placebo gargle on day 1, an ibuprofen gargle on day 2, and an ibuprofen gargle on days 3–5 at least once daily. The primary endpoint of the crossover study period is the change in pain level as measured by a visual analogue scale score from before gargle administration to 5 min after gargle administration on days 1 and 2. The primary endpoint of the long-term extension study is assessment of long-term safety. The results of this study may support existing evidence regarding the effectiveness of ibuprofen rinses in treating OLP.

## 1. Introduction

Oral lichen planus (OLP) is a refractory chronic inflammatory disease, clinically characterized by abnormal keratinization and histopathologically characterized by a series of inflammatory reactions (e.g., a saw-toothed appearance and thickening of the spinous cell layer), which are caused by basal cell damage related to severe lymphocytic infiltration of the subepithelium of the oral mucosa and fluid degeneration at the epithelial–intrinsic layer boundary. Because the cause is unknown and a complete cure is rare, the treatment of OLP is symptomatic [1]. OLP is characterized by lacy white patches on the buccal mucosa, many of which eventually become erythematous lesions with erosions and ulcers. Many patients with OLP experience discomfort and pain in the affected area, which may interfere with their quality of life (QOL), leading to inadequate oral hygiene and further exacerbating inflammation of the oral cavity, thereby creating a vicious cycle [2].

As a symptomatic treatment, steroid-containing ointments are most often recommended for patients with OLP who are experiencing pain. However, steroids have numerous side effects, including secondary candidiasis, nausea, refractory reactions, mucosal atrophy, delayed healing, and systemic absorption, which can result in substantial negative impacts on a patient’s QOL [3]. Other treatments for OLP include the administration of tacrolimus, pimecrolimus, and thalidomide; low-level laser therapy; photodynamic therapy; and surgical resection. All of these treatments were proposed for steroid-resistant patients with OLP; however, none have been comprehensively effective [4]. Therefore, the search for effective drugs against OLP with minimal side effects is an ongoing challenge.

Ibuprofen was developed in the 1960s and is a potent inhibitor of prostaglandin synthesis with antipyretic, analgesic, and anti-inflammatory effects [5]. Because ibuprofen is pharmacologically active against both cyclooxygenase (COX)-1 and COX-2, side effects, including gastrointestinal and renal dysfunction, may occur after systemic administration. However, multiple reviews and meta-analyses have shown that ibuprofen is effective and the least toxic nonsteroidal anti-inflammatory drug (NSAID) for adults and children [6,7]. A previous study demonstrated the effectiveness of an ibuprofen gargle in patients with chemotherapy/chemoradiotherapy-induced oral mucositis [8]. Because ibuprofen gargles undergo rapid local absorption, the present study was designed based on the assumption that ibuprofen gargles are effective for the treatment of potentially painful oral mucosal lesions with erosions and ulcers that appear in patients with OLP.

The primary objective of this study is to estimate the short-term efficacy and long-term safety of an ibuprofen gargle in patients with OLP who are experiencing pain. The secondary objective is to estimate the middle- and long-term efficacy of an ibuprofen gargle in patients with OLP who are experiencing pain.

## 2. Experimental Design

### 2.1. Study Design

This study comprises a crossover study followed by a long-term extension study. The crossover study is a placebo-controlled, double-blind, randomized trial, whereas the long-term extension study is an open-label trial (Figure 1). These studies are being conducted at Kobe University Hospital. The study protocols were registered with the Japan Registry of Clinical Trials (jRCT) (jRCTs051220009, jRCTs051220010).

### 2.2. Patients and Eligibility Criteria

Table 1 and Table 2 list the inclusion and exclusion criteria for the crossover and long-term extension studies, respectively [9].

### 2.3. Gargle Preparation

The ibuprofen gargle will be prepared at the Department of Pharmacy, Kobe University Hospital. A 100 mL volume of gargle will contain 600 mg (0.6%) of ibuprofen, sodium hydroxide, sodium hydrogen carbonate, hydrochloric acid (to adjust pH), glycerin, methyl parahydroxybenzoate, and propyl parahydroxybenzoate. The formulation of the placebo gargle will lack ibuprofen but will otherwise remain identical.

### 2.4. Intervention

Patients in the crossover study period will be equally and randomly assigned to either the ibuprofen–placebo (IP) or the placebo–ibuprofen (PI) groups, using the permutation random block method stratified by category according to the baseline visual analogue scale (VAS) values ranging from 20 to <35 or ≥35 mm (regarded as mild pain and moderate to severe pain, respectively, in our clinic). To ensure blinding, the block size will not be disclosed. The allocation sequence for the randomization method will be generated by a biostatistician, using a random number table to generate a 10-page allocation table. The trial participants, care providers, and outcome assessors will be blinded. Block sizes will be concealed until an analysis of the primary endpoint is completed. The randomization key will be kept secret from data management and statistical analysis personnel throughout the duration of the study, as well as the duration of data bank availability to the public. The randomization key will be held by the study’s third-party allocation manager for the duration of the study.

In the crossover study period, the IP group patients will be administered the ibuprofen gargle on day 1, the placebo gargle on day 2, and the ibuprofen gargle on days 3–5 at least once daily. The PI group patients will be administered the placebo gargle on day 1, the ibuprofen gargle on day 2, and the ibuprofen gargle on days 3–5 at least once daily. This approach will allow assessment of the placebo effect while ensuring adherence to ethical principles by providing the actual drug to all patients with pain during the course of the study. This is a pilot study and no other data are available regarding the frequency of the ibuprofen gargle administration; thus, we will use the minimum frequency of at least once daily. All patients will gargle with approximately 10 mL of the solution, which will be retained in the mouth to ensure its contact with the affected area for ≥30 s (preferably 1 min), and then they will spit out the solution. Previous clinical studies by our group have demonstrated robust safety and efficacy when using these volume and duration parameters [8,9]. The patients will not be allowed to drink water or rinse their mouths for ≥5 min after gargling. Generally, the patients will be allowed an interval of ≥15 min between administrations of the gargle. The maximum daily dosage will be 100 mL of the gargle solution, which is equivalent to 600 mg of ibuprofen (the maximum daily dosage approved for oral intake [10]); this threshold will ensure safety even if a patient swallows the entire volume of the solution each day. Moreover, during the crossover study period, the patients will not be allowed to receive any new treatments for their oral lesions (Table 3).

During the long-term extension study period, all patients will be administered an ibuprofen gargle and will be allowed to receive new treatments for their oral lesions.

Study assessments and the corresponding event schedules are summarized in the assessment schedule presented in Table 4 and Table 5. The timing of assessments in the long-term extension study (Table 5) reflects the standard follow-up procedures in our clinic (visits at intervals of ~2 months).

### 2.5. Data Collection and Management

The investigators will e-mail the coordinators regarding the receipt of a patient enrollment form. Then, the coordinators will check the patient’s eligibility and issue an enrollment confirmation containing the eligibility judgment, randomization assignment result from the generated random sequence, and enrollment number.

The investigators will enter the data from each patient’s medical record into Research Electronic Data Capture (REDCap; an electronic data system for clinical research) to manage the data and protect confidentiality before, during, and after the trial. The principal investigator will confirm that the data in REDCap are complete and correct. Only the biostatistician will have access to the final dataset.

## 3. Outcomes

### 3.1. Primary Endpoint

The primary endpoint of the crossover study period will be to determine the difference in patient pain perception after the ibuprofen gargle and placebo gargle using a VAS ranging from 0 (no pain) to 100 (worst pain) on a continuous scale, immediately before and 5 min after gargling on days 1 and 2 (ΔVAS_5_ibuprofen_ − ΔVAS_5_placebo_) to evaluate the efficacy of the ibuprofen gargle. In the long-term extension study, the primary endpoint will be the assessment of ibuprofen gargle safety, estimated according to the Common Terminology Criteria for Adverse Events version 5.0.

### 3.2. Secondary Endpoints

Secondary endpoints will include determining the difference in patient pain perception after the ibuprofen gargle and placebo gargle using a VAS on a continuous scale, immediately before and 15 min after gargling on days 1 and 2 (ΔVAS_15__ibuprofen − ΔVAS_15__placebo) to evaluate the efficacy of ibuprofen gargle; determining the difference in patient pain perception after the ibuprofen and placebo gargle using VAS on a continuous scale, immediately before and 5 and 15 min after gargling on days 3–5 to evaluate the efficacy of ibuprofen gargle; evaluating the time of onset of ibuprofen’s effect; evaluating the duration of ibuprofen’s effect; determining the change in each domain of the Patient-Reported Oral Mucositis Symptom (PROMS) scale after ibuprofen gargle administration; evaluating the correlation between the overall daily efficacy of ibuprofen gargle and the number of times ibuprofen gargle was administered per day from days 1–5; and estimating the occurrence of adverse events.

## 4. Statistics

### 4.1. Sample Size Calculation

The target sample size is 24, with 12 patients each in the IP and PI groups. In a previous report involving healthy individuals and patients with chemotherapy/chemoradiotherapy-associated oral mucositis [7], the mean ΔVAS ± standard deviation (SD) for pain relief after 3 days of ibuprofen gargling was −12.8 ± 8.4 mm (*n* = 7). In the subgroup with VAS value of ≥30 mm before the ibuprofen gargle, the ΔVAS ± SD after ibuprofen gargle was −15.6 ± 8.1 mm (*n* = 5).

In the present study, the ibuprofen gargle is expected to have a ΔVAS_5_ ± SD of 15.0 ± 3.0 mm among patients with baseline VAS value of 20–34 mm, whereas the placebo gargle is expected to have a placebo effect of 7.5 ± 3.0 mm (i.e., less than half of the ibuprofen gargle ΔVAS_5_ ± SD). Furthermore, the ibuprofen gargle is expected to have a ΔVAS_5_ ± SD of 20.0 ± 10.0 mm among patients with a baseline VAS value of ≥35 mm, whereas the placebo gargle is expected to have a placebo effect of 10.0 ± 10.0 mm (i.e., less than half of the ibuprofen gargle ΔVAS_5_ ± SD). Assuming that the estimated composition ratio of the baseline VAS value is approximately 2:1 (20–34: ≥35 mm), the mean within-subject difference in ΔVAS_5_ (ΔVAS_5_ ibuprofen − ΔVAS_5_ placebo) will be 8.0, the SD will be 7.0–9.0, and the ratio of between- and within-subject variances will be 0.8–1.2. For alpha and beta errors of 0.05 and 0.1, respectively, the number of patients per group would be 4–9, thereby indicating a need for ≥18 patients in total. Additionally, we expect dropout or exclusion of ~25% of patients because of treatment noncompliance or consent withdrawal. Thus, we intend to enroll 24 patients (IP group, *n* = 12; PI group, *n* = 12) in the present study. This sample size calculation was conducted in R software (R Core Team, 2022, version 4.2.0).

### 4.2. Primary Analysis

In the crossover study period, the data will be used to calculate the mean and SD of ΔVAS_5_ on days 1 and 2. Additionally, the 95% confidence interval (CI) of the mean of ΔVAS_5_ will be calculated where appropriate. Furthermore, we will calculate the mean and SD of the within-study ΔVAS_5_ difference, as well as the 95% CI of the mean (where appropriate). Finally, we will assess the treatment effect through the division of the mean of the within-subject ΔVAS_5_ difference by two, followed by the calculation of the *P*-value using Student’s *t*-test. In the long-term extension study period, safety data will be collected, and the 95% CI will be calculated where appropriate. These analyses will be conducted in R software (R Foundation for Statistical Computing).

### 4.3. Secondary Analysis

The mean and SD of ΔVAS_15_ will be calculated on days 1 and 2. Additionally, the 95% CI of the mean of ΔVAS_15_ will be calculated where appropriate. Furthermore, we will calculate the mean and SD of the within-study ΔVAS_15_ difference, as well as the 95% CI of the mean (where appropriate). We will assess the treatment effect through the division of the mean difference of the within-subject ΔVAS_15_ by two, followed by the calculation of the *P*-value using Student’s *t*-test. Summary statistics will be calculated for the time of onset of ibuprofen’s effect at each time point, as well as the duration of its effect at each time point. For each domain of the PROMS scale, a time trend diagram will be created using the least-squares mean, and its 95% CI for each time point will be calculated. Additionally, we will measure the mean of the area under the curve and its 95% CI. We will calculate summary statistics for the overall daily efficacy according to scale, group, and the number of times ibuprofen gargle or placebo gargle was administered. These analyses will be conducted in R software (R Foundation for Statistical Computing).

### 4.4. Data Monitoring

The study will be periodically monitored to ensure that the patients’ human rights and welfare are being protected. The study will be safely conducted in accordance with the protocol and applicable regulatory requirements under the Clinical Trials Act. During data collection, the principal investigator will appoint individuals responsible for overseeing the items specified in the written procedure for study monitoring. To ensure quality assurance, the study will be examined for adherence to the protocol and written procedures, in a manner that is independent and separate from the routine activities of monitoring.

If any questions are raised regarding data summarization or analysis, the biostatistician and the principal investigator will discuss and resolve the issues. In the event of missing data, the investigators will contact the affected patients. If the within-subject difference or ΔVAS cannot be calculated because of missing VAS values, the values will be recorded as zero at the time of analysis. If any missing or deficient values other than VAS are not resolved, no zero values will be used as substitute data.

In this study, an adverse event will be defined as any disease, disability, infection, or death that occurs during the course of the trial. The investigators will record all adverse events in the Case Report Form; affected patients will be treated and followed up until symptom resolution is achieved during the study period. If the investigators detect a potential causal relationship between the adverse events and ibuprofen, all adverse events recorded will be reported to the review board.

## 5. Discussion

To the best of our knowledge, this single-center, placebo-controlled, double-blind, randomized crossover trial will be the first well-designed clinical study to evaluate the efficacy and safety of an ibuprofen gargle in terms of relieving pain caused by OLP and oral lichenoid lesions.

Regarding the safety concerns associated with the ibuprofen gargle, the gargling solution will be retained in the mouth for approximately 1 min, then spit out. Therefore, we presume that systemic absorption of ibuprofen will be unlikely because of this brief contact with the oral cavity. The concentration of ibuprofen used in the present study will be 600 mg ibuprofen/100 mL; thus, the amount of drug that can be accidentally ingested will be less than the maximum daily allowance (600 mg) approved by the Pharmaceuticals and Medical Devices Agency for oral ibuprofen [10]. Accordingly, we presume that adverse events related to the oral absorption of ibuprofen will be less than or equal to the adverse events reported for oral intake of ibuprofen (e.g., prolonged bleeding, jaundice, rash, itch, hives, eczema, purpura, visual abnormality, appetite loss, nausea/vomiting, gastric distress, abdominal pain, and/or edema [10]).

For ethical reasons (i.e., ensuring that all patients with pain receive the actual drug during the course of the study), only ibuprofen gargles will be administered from day 3 onward (Table 4). The present crossover study will be conducted between days 1 and 2 because our previous study showed that the median duration of the effect of the ibuprofen gargle was approximately 20 min, with no carryover effect when comparing days 1 and 2 [8]. Therefore, we selected the ibuprofen gargle as a test drug for the treatment of patients with OLP and oral lichenoid lesions, with a trial efficacy period of 5 days for short-term treatment. This duration was selected to ensure that various degrees of OLP-related pain, which fluctuates over time and throughout the course of a day [11], could be treated with the ibuprofen gargle. Additionally, this duration avoids the potential prolonged use of ineffective medication if the ibuprofen gargle does not result in a clinically significant effect. During the long-term extension study, all patients presenting with adverse events and clinically significant conditions will be followed up until these events and conditions resolve, stabilize, or are judged by the investigator to be no longer clinically significant. Based on our clinical experience, we will perform 6 months of follow-up for safety assessment.

Because previous clinical trials studied only ibuprofen-containing drugs without a placebo/control group, the present clinical trial is designed as a placebo-controlled comparative study to examine the placebo effect [8]. Therefore, this trial aims to evaluate the efficacy of an ibuprofen gargle compared with the efficacy of a placebo gargle for pain relief in patients with OLP and oral lichenoid lesions. Additionally, because OLP is associated with erosions, ulcers, and the loss of keratinized mucosa, topically administered ibuprofen is expected to be rapidly absorbed into the affected area [12].

This study has a few potential benefits. First, because lesions with refractory ulcers (e.g., OLP) are treated with coping strategies, efforts to control oral pain can substantially improve QOL for patients with OLP. However, oral analgesics generally are taken two or three times daily and do not have a long half-life; this leads to intervals of uncontrolled pain. Additionally, pain fluctuates over time. Therefore, gargling as needed (i.e., when pain is present) may allow patients to control symptoms of pain with low doses of ibuprofen while experiencing fewer adverse events compared with oral analgesics. Second, ibuprofen is an inexpensive analgesic; thus, its use is not expected to incur a large financial burden for most patients. Third, this study uses a placebo-controlled, double-blind, randomized crossover design to strengthen the assessment of subjective endpoints (i.e., VAS and PROMS scale). Fourth, this study uses two subjective endpoints: VAS values and PROMS scores (both on continuous scales of 0–100), which serve as indicators of pain intensity and QOL, respectively. The PROMS scale has demonstrated positive correlations with scales that measure depression and negative correlations with scales that measure both physical and emotional well-being [13]; thus, it provides a concise assessment of QOL that has been specifically validated for patients with ulcerative oral lesions. Nevertheless, this study has an important limitation in that it will be conducted at a single center to facilitate the formulation and dispensing of the ibuprofen gargle. Overall, the results of this study may provide valuable evidence regarding the effectiveness of ibuprofen rinses for the treatment of painful OLP.

## 6. Trial Status

This manuscript is based on the current version of the study protocol (version 1.0, last updated on 22 March 2022). The study was first authorized on 22 March 2022. Participant recruitment began on 30 May 2022. The expected date of completion (last visit of the last patient) is 10 April 2024.

## Figures and Tables

**Figure 1 mps-06-00007-f001:**
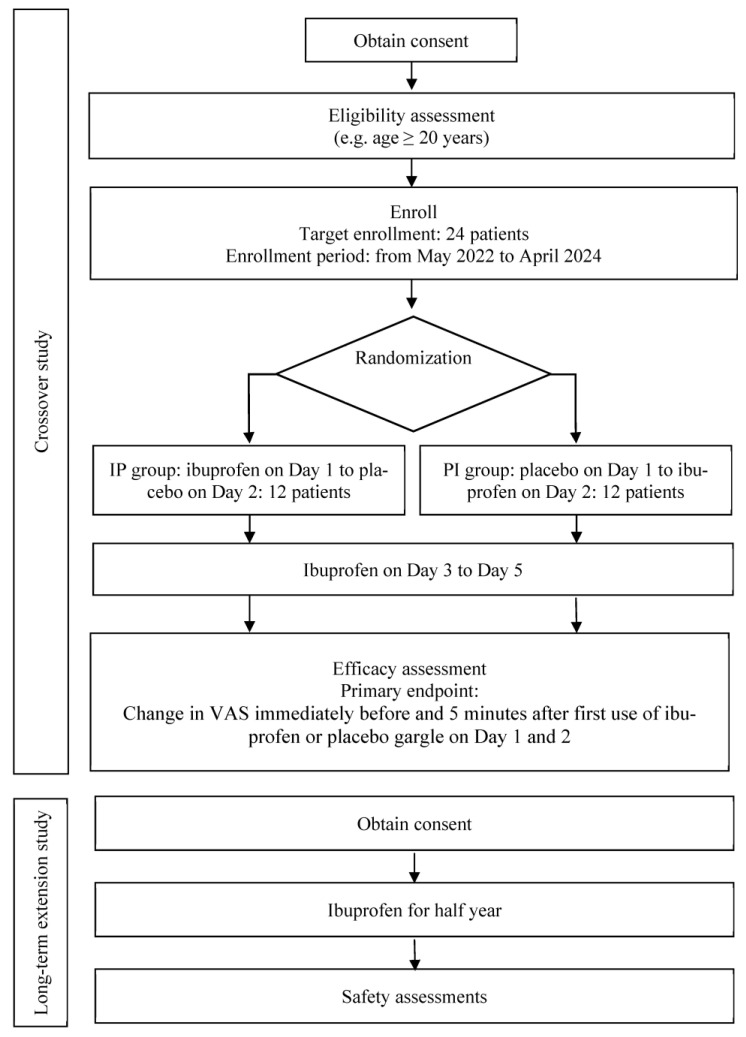
Flowchart of the study design. VAS: visual analogue scale, IP: ibuprofen to placebo, PI: placebo to ibuprofen.

**Table 1 mps-06-00007-t001:** Inclusion and exclusion criteria for the crossover study.

Inclusion Criteria
Patients with oral lichen planus or oral lichen planus-like lesions. ^1^ Patients with pain in the oral cavity averaging ≥20 mm on the Visual Analogue Scale during the 7 days prior to registration. Patients who are receiving treatment (both systemic and local) for oral lesions and have been using the drug at a consistent dose for ≥28 days before the date of registration. Patients aged ≥20 years at the time of obtaining informed consent. Patients from whom documented consent has been obtained regarding their voluntary participation in this clinical study.
**Exclusion Criteria**
Patients with peptic ulcers. Patients with concurrent severe or uncontrolled concomitant medical conditions. Patients with a history of hypersensitivity to any component present in the ibuprofen gargle solution. Patients with impaired cardiac function or clinically significant heart disease. Patients with aspirin-induced asthma. Patients who use analgesic drugs at least once weekly for any chronic pain. Patients with dementia, psychiatric symptoms, drug addiction, or alcoholism. Pregnant or lactating women. Patients considered inappropriate (for miscellaneous reasons) based on the assessment of the investigator or the subinvestigator.

^1^ During the statistical analysis portion of the study, subgroup assessments will be conducted to identify differences in efficacy and safety between patients with oral lichen planus and patients with oral lichen planus-like lesions.

**Table 2 mps-06-00007-t002:** Inclusion and exclusion criteria for the long-term extension study.

Inclusion Criteria
Patients who have completed the double-blind, placebo-controlled, phase II clinical trial of ibuprofen gargle for patients with oral lichen planus or oral lichen planus-like lesions. ^1^ Patients who wish to continue to receive ibuprofen-containing products from the double-blind, placebo-controlled, phase II clinical trial. Patients from whom documented consent has been obtained regarding their voluntary participation in this clinical study.
**Exclusion Criteria**
Patients who met the termination criteria over 2 days for the double-blind, placebo-controlled, phase II clinical trial. Patients who were diagnosed with dementia, psychiatric symptoms, drug addiction, and/or alcoholism, and identified by an attending investigator as neither fully comprehending nor cooperating. Pregnant or lactating women. Patients considered inappropriate (for miscellaneous reasons) based on the assessment of the investigator or the subinvestigator.

^1^ During the statistical analysis portion of the study, subgroup assessments will be conducted to identify differences in efficacy and safety between patients with oral lichen planus and patients with oral lichen planus-like lesions.

**Table 3 mps-06-00007-t003:** New oral lesion interventions that will be prohibited during the crossover study.

Topical Therapy
Rinses and mouthwashes Azulene sulfonate sodium hydrate (Azunol Gargle^®^) Mouthwash containing sodium azulene sulfonate hydrate and sodium bicarbonate (Hachiazule Gargle^®^) Povidone-iodine mouthwash (Isodine Gargle Solution^®^) Benzethonium chloride (Neostelin Green^®^ 0.2% mouthwash solution)
Topical steroids (ointment, spray, patch, mouthwash, topical injection) Triamcinolone acetonide ointment (Kenalog Oral Ointment^®^) Group IV Dexamethasone ointment (Dexaltin Oral Ointment^®^, Aphtasolon Aftazolone Oral Ointment^®^ 0.1%, Dexamethasone Ointment Oral^®^) Group V Beclomethasone propionate spray (Salcoat^®^ Capsule for Oral Spray 50 μg) Group III Hydrocortisone acetate ointment (DESPA^®^ Kowa) Group V Triamcinolone acetonide patch (Aftatch^®^ Adhesive Tablet) Group IV Fluocinonide ointment (Topsym^®^) Group II Dexamethasone solution (Decadron^®^ elixir 0.01%) Triamcinolone acetonide injection (Kenacort^®^) Group IV
Other topical agents Dimethylisopropylazulene ointment (Azunol Ointment^®^) Tacrolimus hydrate ointment (Protopic Ointment^®^) Lidocaine hydrochloride viscous (Xylocaine viscous^®^ 2%)
Excisional surgery or equivalent surgical treatment Surgical excision Cryo (cryosurgery) treatment Laser treatment Argon plasma coagulation therapy High-frequency electrical therapy
**Systemic Therapy**
Cephalantin (CephalantinⓇ) Kampo, the Japanese herbal medicine (Juzentaihoto, Hochuekkito, Bakumondoto, Orengedokuto, Shosaikoto) Vitamin preparations Irsogladine maleate (Gaslon N^®^) Antiallergic drugs (Azeptine^®^, Rizaben^®^) Anxiolytic (Selsyn^®^, Horizon^®^) Anti-inflammatory and analgesic drugs (NSAIDs and acetaminophen) Oral steroid therapy

**Table 4 mps-06-00007-t004:** Summary of the study assessments and procedures in the crossover study.

	Crossover Study						
	Enrollment	Treatment Period					Close-out
	Day 0	Day 1	Day 2	Day 3	Day 4	Day 5	Days 6–10
Enrollment							
Eligibility screen ^1^	×						
Informed consent	×						
Registration	×						
Allocation	×						
**Interventions**							
I then P		×	×				
P then I		×	×				
I				×	×	×	
**Assessments**							
PROMS scale		×				×	
VAS (0 min)		×	×	×	×	×	
VAS (5 min)		×	×	×	×	×	
VAS (15 min)		×	×	×	×	×	
Diary ^2^		×	×	×	×	×	×
Adverse events		×	×	×	×	×	×

^1^ Oral pain (VAS), site of lesion, duration of disease. ^2^ The time of onset of the effect of ibuprofen, duration of the effect of ibuprofen, number of uses (gargles), and self-reported global efficacy. PROMS: Patient-Reported Oral Mucositis Symptom, I: ibuprofen, P: placebo, VAS: visual analogue scale.

**Table 5 mps-06-00007-t005:** Summary of the study assessments and procedures in the long-term extension study.

	Crossover Study	
	Enrollment	Treatment Period	Close-Out
	Day 0	Day 1	Day 55	Day 56, 112	Day 57, 113	Day 111, 167	Day 168
**Enrollment**							
Eligibility screen ^1^	×						
Informed consent	×						
Registration	×						
**Interventions**		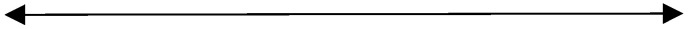	
**Assessments**
PROMS scale		×				×	
VAS (0 min)		×	×	×	×	×	
VAS (5 min)		×	×	×	×	×	
VAS (15 min)		×	×	×	×	×	
Diary ^2^		×	×	×	×	×	×
Adverse events		×	×	×	×	×	×

^1^ Oral pain (VAS), site of lesion, duration of disease. ^2^ The time of onset of the effect of ibuprofen, duration of the effect of ibuprofen, number of uses (gargles), and self-reported global efficacy. PROMS: Patient-Reported Oral Mucositis Symptom, VAS: visual analogue scale.

## Data Availability

Data sharing does not apply to this study protocol because no datasets were generated. However, the data will be made available by the author upon reasonable request when the trial is completed. Please contact the corresponding author for the data generated during this study. The results of the study will be published in a paper or reported in a database (jRCT). All items from the World Health Organization Trial Registration Data Set are available at the Japan Registry of Clinical Trials (jRCT) identifier: jRCTs051220009, registered 30 May 2022 (https://jrct.niph.go.jp/en-latest-detail/jRCTs051220009); jRCTs051220010, registered 30 May 2022 (https://jrct.niph.go.jp/latest-detail/jRCTs051220010).

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
