# Peer review of "Efficacy and Long-Term Safety of Ibuprofen Gargle for Oral Lichen Planus: A Study Protocol of Randomized Crossover and Long-Term Extension Trials"

_mps, 2023, doi:10.3390/mps6010007_

Round 1

Reviewer 1 Report

The papper " Efficacy and long-term safety of ibuprofen gargle for oral lichen planus: a study protocol of randomized crossover and long-term extension trials " , in the actual vertion is a initial research that shold be of great value in the clinical management of OLP. 

1- Consider only OLP cases or at least two groups a) OLP and b) OLP like lesions.

2- The data shold be more clearly presented.

3- Expand the discussion secction.

The above suggestions do not represent the exclusion of the article of the peer review process. 

Author Response

Responses to the comments of Reviewer #1

  • Is the research design appropriate? Can be improved. [from the checklist]

Response: The captions for Tables 1 and 2 have been modified to clarify that subgroup assessments will be conducted to identify differences in efficacy and safety between patients with oral lichen planus and patients with oral lichen planus-like lesions (page 4, lines 86–88 and 91–93).

  • Are the methods adequately described? Can be improved. [from the checklist]

Response: The Experimental Design has been modified to clarify the rationale for VAS thresholds (page 5, lines 104–106), the methodology used for blinding and allocation (page 5, lines 106–114), the names of drugs that will be prohibited during the crossover study (Table 3), and the rationale for timing of assessments in the long-term extension study (page 6, lines 139–141).

  • Are the results clearly presented? Must be improved. [from the checklist]

Response: We respectfully presume that the reviewer is referring to the Statistics section because no Results are included in this Protocol manuscript. We have revised the language throughout the Statistics section to enhance clarity and avoid repetition (page 8, line 187 to page 9, line 255). In the Sample Size Calculation subsection (page 8, lines 188–210), we have rearranged the text and placed less essential information in parentheses, which will allow readers to more fully understand the relationships of numerical expressions within each sentence.

  • Are the conclusions supported by the results? Must be improved. [from the checklist]

Response: We respectfully presume that the reviewer is referring to the rationale and context mentioned in the Discussion section. We have more fully addressed potential safety concerns regarding oral intake of ibuprofen (page 9, lines 261–271). We have addressed ethical considerations influencing our study design, as well as the rationale for the durations of each study within this investigation (page 10, lines 272–287). Finally, we have discussed the strengths and limitations of this study (page 10, lines 295–314).

  • Consider only OLP cases or at least two groups a) OLP and b) OLP like lesions.

Response: As noted in our response to Comment 1, the caption for Table 1 has been modified to clarify that subgroup assessments will be conducted to identify differences in efficacy and safety between patients with oral lichen planus and patients with oral lichen planus-like lesions (page 4, lines 86–88 and 91–93).

  • The data shold be more clearly presented.

Response: As noted in our response to Comment 3, we have revised the language throughout the Statistics section to enhance clarity and avoid repetition (page 8, line 187 to page 9, line 255). In the Sample Size Calculation subsection (page 8, lines 188–210), we have rearranged the text and placed less essential information in parentheses, which will allow readers to more fully understand the relationships of numerical expressions within each sentence.

  • Expand the discussion secction.

Response: We thank the reviewer for the opportunity to more fully explain our rationale and the importance of this study. As noted in our response to Comment 4, we have more fully addressed potential safety concerns regarding oral intake of ibuprofen (page 9, lines 261–271). We have addressed ethical considerations influencing our study design, as well as the rationale for the durations of each study within this investigation (page 10, lines 272–287). Finally, we have discussed the strengths and limitations of this study (page 10, lines 295–314).

Reviewer 2 Report

Introduction

- Authors are requested to proofread the initial part of the introduction; I suppose the word "nail" is a typo (line 30)

Experimental Design

- Please fix the position of Figure 1.

- No mention is made in the text regarding the dose-ranging of ibuprofen used in gargles. How was the optimal dose selected? How was the quantity of liquid indicated? How was the time of gargling preferred? What are the safety/efficacy criteria for the considered selected dose of this protocol? I hope that the authors will deepen these aspects in the text.

- Check the correctness of Tables 4-5 and their explanatory captions. Why were brackets used to indicate VAS during the treatment period? Is this explanation worth going into the text?

- Please check the verb times used in each paragraph and use the correct ones.

Author Response

  • English language and style are fine/minor spell check required. [from the checklist]

Response: As suggested, the manuscript has been carefully reviewed by an experienced editor whose first language is English and who specializes in editing papers written by researchers whose native language is not English.

  • Are the methods adequately described? Can be improved. [from the checklist]

Response: As indicated in our response to Comment 2 from Reviewer 1, the Experimental Design has been modified to clarify the rationale for VAS thresholds (page 5, lines 104–106), the methodology used for blinding and allocation (page 5, lines 106–114), the names of drugs that will be prohibited during the crossover study (Table 3), and the rationale for timing of assessments in the long-term extension study (page 6, lines 139–141).

  • Authors are requested to proofread the initial part of the introduction; I suppose the word "nail" is a typo (line 30)

Response: As suggested, the Introduction has been revised to avoid suggesting that nails are involved in OLP (page 1, lines 30–31).

  • Please fix the position of Figure 1.

Response: As requested, Figure 1 has been moved into the main text (page 3). The figure has also been modified to clearly delineate the crossover study and long-term extension study.

  • No mention is made in the text regarding the dose-ranging of ibuprofen used in gargles. How was the optimal dose selected? How was the quantity of liquid indicated? How was the time of gargling preferred? What are the safety/efficacy criteria for the considered selected dose of this protocol? I hope that the authors will deepen these aspects in the text.

Response: We thank the reviewer for these important questions. We have clarified the composition of the gargle solution, although the chemical ratios are currently proprietary information (page 5, lines 97–99). We have explained that the dose, duration, and volume of ibuprofen gargle used in this study previously exhibited robust safety and efficacy in clinical studies by our group (page 5, lines 125–126)—our safety and efficacy criteria were established in the previous studies. Additionally, we have explained our rationale for using a crossover trial design, as well as the minimum frequency of at least once daily (page 5, lines 118–122). We have also explained our rationale for maximum daily dosage (page 5, lines 129–131).

  • Check the correctness of Tables 4-5 and their explanatory captions. Why were brackets used to indicate VAS during the treatment period? Is this explanation worth going into in the text?

Response: As suggested, Tables 4 and 5 have been carefully checked for accuracy and clarity. The superscript “1” has been added after “VAS” for the 5 min and 15 min assessments in both tables. Additionally, the presence of parentheses was not associated with any specific meaning, so the parentheses have been removed from both tables. Finally, the rationale for timing of assessments in the long-term extension study (shown in Table 5) has been explained in the main text (page 6, lines 139–141).

  • Please check the verb times used in each paragraph and use the correct ones.

Response: As suggested, verb tenses have been corrected in all parts of the manuscript.